**Data Availability Statement:** Data from the Healthy Start intervention contains sensitive information and cannot be made publicly available

# Effects of the healthy start randomized intervention on psychological stress and sleep habits among obesity-susceptible healthy weight children and their parents

Nanna Julie Olsen[ID][1]*, Sofus Christian Larsen[1], Jeanett Friis Rohde[1], Maria Stougaard[2], Mina Nicole Händel[1], Ina Olmer Specht[1], Berit Lilienthal Heitmann[1,3,4]

1 Research Unit for Dietary Studies, The Parker Institute, Bispebjerg and Frederiksberg Hospital, the Capital Region, Denmark, 2 Department of Psychology, Center for Early Intervention and Family Studies, University of Copenhagen, Copenhagen K, Denmark, 3 The Boden Group, Faculty of Medicine and Health, Sydney University, Sydney, Australia, 4 Department of Public Health, Section for General Practice, University of Copenhagen, Copenhagen, Denmark

* nanna.julie.olsen@regionh.dk

## Abstract

Poor sleep and psychological stress are obesity determinants that are rarely included in obesity prevention programs. The aim was to report the effects of the Healthy Start randomized intervention on the secondary outcomes psychological stress and sleep duration and onset latency. Data was obtained from the Healthy Start randomized intervention conducted in 2009–2012 among Danish healthy weight children aged 2–6 years, who had either a high birth weight (>4,000 g), high maternal pre-pregnancy body mass index (>28 kg/m$^2$), or low maternal educational level (≤10 years of schooling) and their parents. The intervention was designed to deliver improvements in diet and physical activity habits, optimization of sleep habits, and reduction of psychological family stress. The average intervention period was 15 months. Children with information on a 7-day sleep record, sleep onset latency, Strengths and Difficulties Questionnaire (SDQ), and a modified version of Parenting Stress Index (PSI) were included. The effects of the intervention on sleep habits, PSI scores, SDQ Total Difficulties (SDQ-TD) and Pro-social Behavior scores, and 95% Confidence Intervals (95% CI) were analyzed using linear regression intention-to-treat (n = 543 (intervention group n = 271, control group n = 272)) analyses. No statistically significant effects on sleep duration, sleep onset latency, PSI score, or SDQ Pro-social Behavior score were observed. Values both before and after the intervention were within the normal range both for children in the intervention and children in the control group. Mean change in SDQ-TD was 0.09 points (95% CI -0.57;0.59) in the intervention group, and -0.69 points (95% CI -1.16; -0.23) in the control group (p = 0.06). In conclusion, there were no intervention effects in relation to sleep duration, sleep onset latency, PSI score, or SDQ Pro-social behavior. There was an indication that children in the intervention group had slightly more behavioral problems than the control group after the intervention, but values were within normal range both before and after the intervention, and the difference is not considered to be clinically meaningful.

for ethical and legal reasons. Public availability may compromise participant privacy, and this would not comply with Danish legislation (www.datatilsynet. dk). Access to the data requires an application submitted to and subsequently approved by the steering committee. Data requests may be sent to Professor Berit L. Heitmann (Berit.Lilienthal. Heitmann@regionh.dk) or the Research Unit for Dietary Studies at The Parker Institute (bfh-eek@regionh.dk).

**Funding:** The Healthy Start intervention was funded by grants from the Danish Medical Research Council (grant number 271-07-0281), TrygFonden (grant number 7984-07), and the Danish Health Foundation (grant number 2008B101), all awarded to BLH. The Parker Institute at Bispebjerg and Frederiksberg Hospital was supported by a core grant from the Oak Foundation (OCAY-18-774-OFIL). The funders had no role in study design, data collection and analysis, decision to publish, or preparation of the manuscript.

**Competing interests:** The authors have declared that no competing interests exist.

# Introduction

Although children are not commonly identified as susceptible to stress, chronic exposure to stressful situations in the environment is common [1]. As chronic stress can initiate inflammatory processes in the body, which can be expressed e.g. in adipose tissue, muscle mass and hormones, stress may have adverse implications for children's health, and may lead to an increased risk of obesity, metabolic syndrome, and cardiovascular disease [2, 3]. Information on stress in children may be obtained by measuring reactivity to stressors, for example as behavior problems. This approach is supported by previous research; in a subsample from a larger longitudinal study in the US, Hypothalamic Pituitary Axis (HPA) reactivity was measured at age 7, and internalizing symptoms via teachers reports were measured at age 5 and 11. The study found that greater HPA reactivity at age 7 was associated with greater increases in internalizing symptoms between age 5 to 11 years [4]. Similarly, a study embedded in the Dutch "Generation R" cohort found that variations in diurnal cortisol rhythm measured at age 12–20 months were associated with change in internalizing problems between 1.5 and 3 years [5].

Young children are dependent on the care of their parents, and parenting stress has been bidirectionally linked with child stress [6]. A study published in 2019, among 835 parent-child dyads, found that family conflict mediated the association between children's behavior problems at age 1 and parenting stress at age 3, while the association between parenting stress at age 1 and behavior problems at age 3 was mediated by parental supportiveness [6]. Likewise, a recent longitudinal study found that household chaos during preschool predicted a more blunted diurnal cortisol slope in middle childhood, and that greater negative life events and greater concurrent family conflict were associated with increased free cortisol reactivity in middle childhood [7]. This suggests that efforts to reduce psychological stress in young children should focus on the entire family. Intervention tools that have been applied in previous studies to reduce psychological stress in both children and parents include increasing family time [8], resilience capacities enhancement [9], parenting training/education [10–12], and mindfulness for parents [13].

Up to 50% of parents report perceived sleep problems in toddlers, including night waking and bedtime resistance [14]. In addition to impacting daily social-emotional functioning [15], poor sleep in toddlers and primary school children can impact child growth via increased cortisol and decreased melatonin levels [16]. Because sleep problems like sleep consolidation and sleep regulation involve elements of learned behavior by definition, they are suggested to be amenable by behavioral strategies [14]. The most prevalent components in behavioral interventions targeting sleep improvements are sleep education and sleep hygiene [17]. While sleep education provides recommended sleep guidelines and emphasizes the importance of sufficient sleep in relation to health and cognition, sleep hygiene provides practices intended to support optimal sleep health [17]. In children, sleep hygiene include regular bedtime routines, an environment that supports sleep, relaxation exercises, and avoiding stimulating activities [17–19].

The aim of this study was to evaluate the effects of a multifaceted intervention on child behavioral problems, parental psychological stress, as well as on sleep duration and sleep onset latency. The intervention focused on improving diet, increasing physical activity, improving sleep duration and quality, and reducing psychological stress in the family among young children susceptible to overweight and obesity. Previously, we have reported small intervention effects on body composition (prespecified primary outcome) [20], and on diet and physical activity (secondary outcomes) [21, 22].

# Methods and materials

Based on information from the Danish Medical Birth Registry (DMBR) [23], all children born in 11 municipalities in the greater Copenhagen area between 01.01.2004 and 31.12.2007 with either a

high birth weight (> 4.000 g.), or a mother with overweight prior to pregnancy (BMI >28 kg/m$^2$) (measured at the first visit at the general practitioner after becoming pregnant) were selected. Based on information from the administrative birth forms, children (n = 378) with maternal low educational level ($\leq$ 10 years) were also selected in one municipality. In total, 5,902 children, aged 2–6 years at the time of accessing the DMBR, were selected. After selection of the study population, each child was assigned a five- or six-digit project identification number and sorted in random order. Afterwards, selected children were randomized using computer-generated randomization stratified on municipality and with simple randomization at family level. The random allocation sequence was generated by the project's data manager. The project staff enrolled participants and assigned the children to the intervention or the control groups. They also delivered the intervention, and therefore could not be blinded regarding the group assignment of each child.

Participants were randomized into an intervention group (40%), a control group (40%), and a shadow control group (20%). Information on stress and sleep was not available in children in the shadow control group, and consequently this group will not be described further.

Children who i) moved to another municipality after birth, ii) had requested protection from participation in statistical or scientific surveys based on data delivered from the Danish Central Person Registry, iii) had no permanent address, iv) lived in a children's home, v) had died or emigrated, or vi) were registered in the Danish Central Person Registry as being disappeared or had unknown life status, were excluded (n = 2,180).

At baseline, children in the intervention group (n = 320) had height and weight measured, and BMI was calculated. The International Obesity Task Force (IOTF) criteria for overweight (including obesity) according to age and gender from 2000 was applied [24]. Children with overweight at baseline (n = 49) were excluded, while children with healthy weight (n = 271) were included. The median number of consultations was 4 (range 2–7). The intervention period lasted on average 1.3 years (15 months). A total of 161 children completed the study.

Children in the control group (n = 315) had height and weight measured at baseline, and BMI was calculated. Children with overweight were excluded (n = 43), and children with healthy weight were included (n = 272). After on average 1.3 years, the control children and their families were invited to a follow-up examination. A total of 204 children completed the study.

**Fig 1** shows a flow diagram of the Healthy Start study.

The Healthy Start intervention was conducted between May 1st, 2009 and August 31st, 2012. Trial duration was decided a priori. Deviations from the original study protocol (S1 Appendix) comprise inclusion of children from additional 6 municipalities and lowering BMI cut-off for maternal overweight prior to pregnancy from 30 kg/m$^2$ to 28 kg/m$^2$, to increase the sample size. The intervention period was shortened from the expected 2 years to 1.3 years, because the planning of practicalities related to the conduction of the study and recruitment of participants took longer than anticipated. The Healthy Start study is registered at Clinical-Trials.gov (**ID** NCT01583335).

## The intervention

The intervention delivered guidance tailored to each of the families on how to optimize diet and increase physical activity level, as well as reduce stress in the family, and improve sleep quality and quantity of the child. Consultations took place in locations that were provided by the municipalities (e.g. schools), to ensure short transportation time for the participants.

The framework for the counselling process was based on the stages of change principles and on motivational interviewing [25, 26].

The sleep component of the intervention was focused on sleep hygiene, and recommendations for sleep duration. The stress component of the intervention was focused on spending

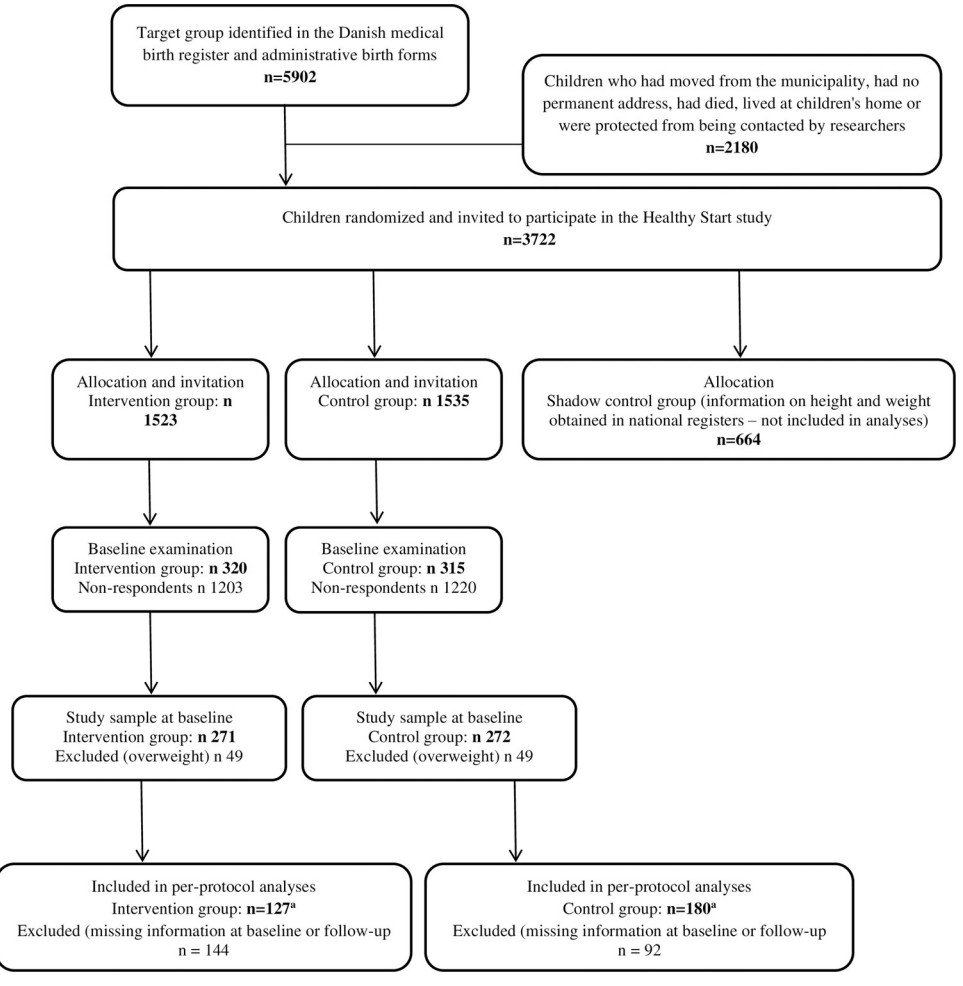

**Fig 1. Flow chart.**

more time as a family. The health consultants planned each consultation using developed key-words focusing on *what* to do to acquire/maintain healthy habits for each of the four intervention areas.

Tools to change behavior in a healthier direction were created for each keyword, to assist the consultant in providing suggestions to the family on *how* to do it. Examples of tools include "Provide clear rules and routines around bedtime" (sleep hygiene), or "Build up every-day routines" (stress). Together with the consultant, each family decided which tools they were motivated to work with until the next consultation, using the stages of change model [25]. To support the health consultants, and to ensure homogeneity in the delivery of the intervention, detailed descriptions on how to provide the consultations were developed by the health consultants in collaboration with the project management. The content of each session was selected from evidence-based determinants of the lifestyle habits that the Healthy Start intervention study aimed to optimize. For example, a high media consumption has been associated with a shorter sleep duration [27], and lowering media consumption could therefore increase the sleep duration. The delivery of the sessions was pilot-tested and trained internally by the health consultants and the project management. Detailed descriptions on how each session was provided have been published previously [20].

Consultations were supplemented by written information in the form of pamphlets on sleep and stress that were developed by project staff and given to the parents. The baseline session lasted for approximately one hour, while the follow-up consultation lasted for approximately one and a half hour. All sessions between baseline and follow-up lasted for approximately half an hour. The time gap between sessions was set to be maximum 4 months, corresponding to at least 3 sessions during the intervention period.

The Healthy Start intervention has been described in more details previously [20, 28].

## Ethics

The Scientific Ethical Committee of the Capital Region in Denmark decided that the project was not a bioethics project, and consequently did not need approval from the Danish Bioethics Committee (journal number H-A-2007-0019). The Danish Data Protection Agency approved the study (journal number: 2015-41-3937). Written informed consent to use the collected data for research purpose was obtained from all parents.

## Outcomes

Parents in both the intervention group and the control group were asked to complete a parental questionnaire at baseline and follow-up. Information about which parent (father/mother) that completed the questionnaire was not obtained. Prespecified outcome measures were child stress, parental stress, sleep duration, and sleep onset latency, which are secondary outcomes of the Healthy Start intervention.

## Child behavioral problems

To indicate child psychological stress, the Danish single-sided version of the Strengths and Difficulties Questionnaire (SDQ) was included in the parental questionnaire.

The SDQ asks about 25 attributes, some positive and others negative. These 25 items are divided into 5 scales ("Emotional symptoms", "Conduct problems", "Hyperactivity/inattention", "Peer relationship problems", and "Prosocial behavior"). The scores of the "Emotional symptoms", "Conduct problems", "Hyperactivity/inattention", and "Peer relationship problems" scales are summed to a SDQ-TD score, based on a scoring syntax available from the SDQ-webpage [29], and used as an exposure variable. The SDQ-TD score ranges from 0–40 points, with higher scores being worse.

The score of the "Prosocial behavior" scale (SDQ-PSB score) is not incorporated into the SDQ-TD score, as absence of prosocial behaviors differs conceptually from the presence of psychological difficulties [30]. The SDQ-PSB score ranges from 0–10 points, with higher scores being best.

The SDQ has been completed for nearly 100,000 children and adolescents in both population studies and clinical samples in the Scandinavian countries [31]. Moreover, in four large-scale Danish cohorts, SDQ has shown to be a useful screening tool for boys and girls across age groups (between 5 and 12 years) and raters (parents and teachers) [32].

## Parenting stress

The Parenting Stress Index (PSI) is a self-report inventory designed to measure parental experiences of stress in the parent-child relation [33]. In order to assess the parental perceptions of the family's well-being in terms of overall stress, 10 out of 32 questions were selected from the Swedish version and modified according to context. The questions asked which changes in life the parents had perceived since they had had the child, regarding 10 dimensions: sleep, stress,

worries, time for themselves, household conflicts, work load, social gatherings in the home, joy of life, everyday surplus energy, and complexity of being a parent compared to expectations. The response options to the questions were "more", "less", or "no difference" compared to before having the child (e.g. "more sleep", "less sleep", or "no difference in sleep"). S1 Table shows the selected questions (translated from Danish into English). Each question was scored between 0 and 2 (with 0 being the best score and 2 being the worst), according to its estimated indication of an overall stress level. The PSI score ranges from 0–20, with higher scores being worse.

The Swedish version of the PSI has been validated for measuring experienced parental stress in mothers of young children [33].

## Sleep habits

The parental questionnaire included a 7-day sleep record, information on duration of sleep onset latency, as well as short questions related to sleep quality. Parents were asked to record the exact times that their child fell asleep in the evening and woke up in the morning from Monday to Sunday to capture nighttime sleep duration on both weekdays and weekend days. Average nighttime sleep duration was calculated as the mean of 6 nights (i.e. Monday evening to Tuesday morning, Tuesday evening to Wednesday morning, etc.) in hours at baseline. In addition, parents were asked to report the number of minutes usually spent from the child was put to bed until sleeping (sleep onset latency).

## Covariates for multiple imputations

Information on native municipality of the child, sex and age, was obtained from the DMBR [23].

In the parental questionnaire, both parents individually reported their highest educational level with nine response options, of which 8 response options were regrouped into three categories: 1) low education level ("primary and lower secondary", "upper secondary", "one or more short courses" and "skilled worker"), 2) medium education level ("short-term further education [<3 years]" or "medium-term further education [3–4 years]"), and 3) High education level ("long-term further education [>4 years]", "research level"). It was not considered possible to include the ninth response option ("other", n = 13) in any of the regrouped categories.

To indicate the children's physical activity level, parents were asked"*How physically active is the child compared to other children at the same age*". The parents could answer if they perceived their child as being *"not so active"*, *"fairly active"*, *"very active"* or *"don't know"*.

Information on parent's perception of their child's sleep quality was based on the question *"How do you perceive your child's sleep*?*"*. The response options were *"the child sleeps calm through the night"*, *"the child sleeps a little disturbed"*, *"the child sleeps disturbed and wakes up once in a while"*, and *"the child sleeps very disturbed and wakes up several times during the night"*. Parents were also asked whether the child takes naps during the day ("yes" or "no").

## Statistical methods

The sample size of the Healthy Start Study was established with the main purpose of having enough statistical power to detect a potential effect of the intervention on the primary outcome (body weight). As the present results represent a secondary study, the sample size was not determined for the purpose of these analyses. However, minimal detectable effects are calculated for all outcomes included in the present study (S2 Table), as results may not be useful without knowing if meaningful effects could be detected [34].

To test the effect of the intervention, linear regression models with treatment status included as the explanatory variable and changes in sleep duration, sleep onset latency, SDQ scores or PSI during follow-up included as the response variables were conducted. All models were adjusted for baseline measure of outcome. Model assumptions (consistency with a normal distribution and variance homogeneity) were assessed for all models through normal probability plots and residual plots.

**Modified intention-to-treat analyses.** Children who dropped out between baseline and follow-up were included in the analyses according to the intention-to-treat (ITT) principle using multiple imputation. In the multiple imputations, m = 10 complete data sets were generated. In each set of data, the missing values were replaced with imputed values, constructed based on predictive distributions for each of the missing values. Each of the completed data sets were analyzed, and the results from the ten analyses were combined to create a single set of estimates that comprised the variability associated with the missing values. The imputations were made using chained equations as implemented in Stata through the commands *ice* and *mim*, and based on the variables described above. Imputations were based on allocated group, municipality of birth, baseline BMI z-score, sex, age at baseline, maternal and paternal educational level, perception of the child's sleep, if the child takes naps during the day, and physical activity compared to children in the same age. When average nighttime sleep duration was used as outcome, imputations were also made on baseline average nighttime sleep duration and SDQ-TD score. When sleep onset latency was used as outcome, imputations were also made on baseline sleep onset latency and SDQ-TD score. When SDQ-TD score was used as outcomes, imputations were also made on baseline SDQ-TD score and average nighttime sleep duration. When SDQ-PSB score was used as outcome, imputations were also made on baseline SDQ-PSB score and average nighttime sleep duration. When PSI score was used as outcome, imputations were also made on baseline PSI score.

The number of missing values ranged from 0 for the variables allocated group, municipality of birth, baseline BMI z-score, sex, and age at baseline, to 239 for paternal educational level. Distribution of observed and missing values of each variable included in the imputations are shown in S3 Table.

All statistical tests were two-sided with a significance level at 0.05. Analyses were performed using Stata SE 14 (StataCorp LP, College Station, Texas, USA; www.stata.com).

**Sensitivity analyses.** Per protocol analyses, removing data from children who dropped out of the intervention before the 1.3 years follow-up examination, were performed as sensitivity analyses. Per protocol analyses are shown in S4 Table. In addition, possible effect modification by sex was examined for all outcomes by adding product terms to the models. Subgroup analyses were conducted if statistically significant interactions were observed. S5 Table shows the specific outcomes stratified by completers and non-completers. Non-completers had a statistically significant higher SDQ-TD score than completers, while no other significant differences were observed.

## Results

Baseline characteristics of the children included in the analyses are presented in **Table 1**. No differences in baseline characteristics were observed between the two groups.

Sample size was n = 543. Sensitivity analyses showed essentially similar results as the ITT analyses presented below (S4 Table). We found no evidence of effect modification by sex for any of the outcomes (all P> 0.102).

### Child behavioral problems

The mean change in SDQ-TD was 0.09 points (95% CI -0.57;0.59) in the intervention group, and -0.69 points in the control group (95% CI -1.16; -0.23) (p = 0.06) (**Table 2**).

**Table 1. Baseline characteristics of the included participants stratified by intervention status[1].**

| | Intervention group | | Control group | |
|---|---|---|---|---|
| | n | Mean (SD) | n | Mean (SD) |
| **Age (years)** | 271 | *4.02 (1.08)* | 272 | *4.02 (1.07)* |
| **Gender (% boys)** | 271 | *55.35* | 272 | *61.03* |
| **BMI Z-score (SD)** | 271 | *0.06 (0.80)* | 272 | *0.15 (0.74)* |
| **Duration of sleep (hours)** | 251 | *10.72 (0.61)* | 253 | *10.75 (0.63)* |
| **Sleep onset latency (minutes)** | 250 | *18.34 (14.51)* | 256 | *18.60 (13.85)* |
| **SDQ Total Difficulties score[a] (points)** | 253 | *7.06 (3.87)* | 256 | *6.36 (3.97)* |
| **SDQ Prosocial Behavior[b] (points)** | 253 | *7.67 (1.86)* | 256 | *7.81 (1.76)* |
| **Parenting Stress Index[c] (points)** | 236 | *13.59 (2.76)* | 236 | *13.53 (2.34)* |
| **Sleep perception[d]** | 254 | | 257 | |
| Calm (%) | 153 | *60.24* | 160 | *62.26* |
| A little disturbed (%) | 72 | *28.35* | 74 | *28.79* |
| Disturbed (%) | 24 | *9.45* | 21 | *8.17* |
| Very disturbed (%) | 5 | *1.97* | 2 | *0.78* |
| **Afternoon sleep (% yes)** | 255 | *32.55* | 257 | *31.91* |
| **Physically active[e]** | 253 | | 257 | |
| Not so active (%) | 3 | *1.19* | 2 | *0.78* |
| Fairly active (%) | 92 | *36.36* | 97 | *37.74* |
| Very active (%) | 154 | *60.87* | 155 | *60.31* |
| Don't know (%) | 4 | *1.58* | 3 | *1.17* |
| **Maternal education (% high level)** | 128 | *24.22* | 186 | *24.73* |

[1]: Results presented as mean (Standard Deviation) unless otherwise stated.

[a]: Range 0–40, higher scores are worse

[b]: Range 0–10, higher scores are better

[c]: Range 0–20, higher scores are worse

[d]: Based on the question "How do you perceive your child's sleep?"

[e]: Based on the question "How physically active is the child compared to other children at the same age?"

SDQ: Strengths and Difficulties Questionnaire.

The mean change in SDQ-PSB score was 0.47 (95% CI 0.17;0.76) in the intervention group, and 0.38 (0.19;0.58) in the control group (p = 0.61) (**Table 2**).

## Parenting stress

Mean changes in PSI score were 2.27 points (95% CI 1.89;2.66) in the intervention group, and 2.09 points (95% CI 1.72;2.47) in the control group (P = 0.46) (**Table 2**).

## Sleep habits

Mean changes in average nighttime sleep duration were -0.01 hours (corresponding to 0.6 minutes) (95% CI -0.11;0.09) in the intervention group, and 0.04 hours (corresponding to 2.4 minutes) (95% CI 0.06;0.08) in the control group (p = 0.71) (**Table 2**).

Mean changes in sleep onset latency were -1.23 minutes (95% CI -3.12;0.66) in the intervention group, and -2.87 minutes (95% CI -4.56; -1.19) in the control group (p = 0.13) (**Table 2**).

## Discussion

In this multifaceted family-based randomized intervention among healthy weight children with obesity susceptibility, we did not observe an effect of the intervention on sleep duration

**Table 2. Changes in sleep and stress between baseline and follow-up (1.3 years).**

| | n | Intervention group Mean change (95% CI) | Control group Mean change (95% CI) | P-value |
|---|---|---|---|---|
| Duration of sleep (hours)[1,2] | 543 | -0.01 (-0.11; 0.09) | 0.04 (0.06; 0.08) | 0.71 |
| Onset latency (minutes)[1,2] | 543 | -1.23 (-3.12; 0.66) | -2.87 (-4.56; -1.19) | 0.13 |
| SDQ Total Difficulties score (points)[1,3] | 543 | 0.09 (-0.57; 0.59) | -0.69 (-1.16; -0.23) | 0.06 |
| SDQ Prosocial Behavior (points)[1,3] | 543 | 0.47 (0.17; 0.76) | 0.38 (0.19; 0.58) | 0.61 |
| Parenting Stress Index (points)[1,3] | 543 | 2.27 (1.89; 2.66) | 2.09 (1.72; 2.47) | 0.46 |

SDQ: Strengths and Difficulties Questionnaire

[1]: Imputations on municipality, gender, age, parental education, physical activity and baseline values of each separate outcome.

[2]: Imputations on parental perception of child sleep quality, taking naps during the day, baseline SDQ-TD

[3]: Imputations on average nighttime sleep duration at baseline.

Difference between groups tested using linear regression modeling with information on outcome, group allocation, and baseline measure of outcome. Results presented as mean changes (95% CI).

or sleep onset latency. Likewise, we did not observe an effect of the intervention on parenting stress. In contrast, we observed that the intervention group had a higher score as well as a larger score change in SDQ-TD, corresponding to lower socio-emotional skills or more psychological problems.

The lack of significant differences in sleep duration and sleep onset latency after the intervention may be because children in our sample had both a sleep duration and sleep onset latency within a range that can be considered as normal [35], which left little or no possibilities for improvement.

Because sleep problems may have different trajectories between infancy and middle childhood, as investigated by Williamson and colleagues [36], another possible explanation for our results could be that the follow-up period was too short to capture a prevention effect. However, the post hoc power calculations of minimal detectable effects in this study does not suggest that we have missed any clinically relevant intervention effects.

The observations of no significant differences in sleep duration and sleep onset latency after the intervention are similar to observations in previous studies; the IDEFICS study was conducted across 8 European countries (Belgium, Cyprus, Estonia, Germany, Hungary, Italy, Spain and Sweden) among 2–9 year old children, and implemented a multilevel intervention including sleep duration as a key behavioral target [37]. Although the authors observed a smaller decrease in weeknight sleep duration over the 2 years follow-up period in the intervention group compared to the control group, they concluded that the sleep component of the intervention did not lead to clinically relevant changes in sleep duration [37]. Similarly, a large cluster RCT of 3713 Chinese children introducing a school-based sleep education program on sleep duration among adolescents did not show any effects [38]. Finally, follow-up data from a parallel, 4-arm, single-blind, 2-year randomized controlled trial did not show any significant differences in nighttime sleep duration, even though results suggested that the trial arm with a brief sleep intervention in infancy reduced the risk of obesity at age 2, 3.5 and 5 years compared to the other three trial arms [39]. Considering that short sleep duration has been linked to development of obesity in children in observational studies [40], these results suggest that development of programs that are effective in improving children's sleep habits are urgently needed.

Possible explanations for the observed non-significant differences may include the risk of type 2-error, selection bias (reflecting that the study population generally had a high socio-economic position), or attrition bias. The intervention group had a larger drop-out rate than the

control group, which may have introduced a systematic drop-out that is not fully adjusted for in the multiple imputations. Even though we only observed a statistically significant difference between completers and non-completers in SDQ-TD score, this cannot be ruled out as a possible reason for failure to find effects. Another possible explanation may be that the intervention comprised multiple components rather than a single component. Even though this is a strength in terms of obesity being multifactorial, it may be hypothesized that addressing multiple lifestyle areas within a given intervention time period could reduce the intervention dose that is delivered for a specific lifestyle area, which may attenuate the exposure degree between the intervention and the control groups. Finally, the information given to all families, including those in the control group, e.g. that their child was susceptible to develop overweight and obesity, may have led to control group families having self-facilitated changes in for example physical activity level which could impact sleep habits [41]. Based on the minimal detectable effects from the power calculations for these secondary outcomes, it is considered unlikely that we have missed intervention effects that would be clinically relevant.

We did not observe any effects of the intervention on parenting stress. This implies that the intervention was not successful in reducing parenting stress (or that parents in our sample were not sufficiently stressed and that there consequently was little to intervene on), but also that participating in the intervention did not introduce additional stress to the parents. This is in line with results from a recent Canadian study that found neutral effects of their intervention on family stress level [42]. Considering that parent stress has been associated with lower physical activity and higher sedentary behaviour among the children [43], home food environment and dietary patterns [44], as well as child weight status [45], future obesity prevention interventions are still encouraged to include parent stress as a focus point in the program development. The most important limitation to our results on parenting stress is that the instrument was modified according to context and consequently was not validated, which may have increased the risk of type 2-error.

Our results suggested that children in the intervention group had increased their SDQ-TD score after the intervention period, hence had a higher degree of behavioural problems. This was surprising but may be a chance result due to multiple testing. In this regard, the SDQ-TD scores were within normal range at both baseline and follow-up, and the size of the increase in SDQ-TD score in the intervention group is not considered to be clinically significant. To the best of our knowledge, no previous obesity prevention interventions have applied SDQ as outcome measure, and it is therefore not possible to directly compare our results to previous findings. However, one previous intervention to improve emotional and behavioural self-regulation in combination with an obesity prevention program in 3–4 year old children found improved teacher-reported self-regulation, measured by a modified 60-item version of the Social Competence and Behavior Evaluation [46]. Potentially, as the intervention was multi-component, the stress component of the Healthy Start intervention was not developed or delivered sufficiently during the intervention to be effective. Another possible explanation for the observation in our study could be that participating in the intervention that focused on behaviour change have been more demanding and straining and may have affected children in the intervention group adversely in relation to stress.

Strengths of the Healthy Start study include the randomized design, which reduces the risk of potential confounding. However, the health consultants were not blinded and were therefore aware of the group allocation of the children which may have introduced a risk of observation bias. To minimize the risk of observation bias, detailed manuals and guidelines on how the consultations with both the intervention and control groups were to be conducted were developed [20].

The Healthy Start study also has limitations; the study included self-reported information that may introduce misreporting. Information on whether the sleep diary was completed by

the same parent putting the child to bed in the evening and taking the child out of bed in the morning, and on which parent completed the SDQ was not obtained, which may have introduced some non-differential misclassification. Likewise, we do not have information on whether the same parent completed the PSI questionnaire at baseline and follow-up. This may have introduced some misclassification, because perceived stress may differ between mothers and fathers, as explored (for a different family stress scale), by Cooke and colleagues [47]. The consequence is an attenuation of the observed differences. In our study, child behavioral problems were used as an indicator of child psychological stress. Even though previous studies have found associations between behavioral problems and cortisol [48–52], behavioral problems could also be related to factors such as infant temperament [53] and parenting skills [54]. Another limitation to our study is hence that the extent to which SDQ-TD scores reflect stress in the child remains an open question.

Interpretation of the results may only be generalized with caution and may not be applicable to healthy weight children without obesity susceptibility. Finally, socioeconomic position may impact the willingness to participate in the Healthy Start study, which could introduce selection bias and could also reduce the generalizability of the results.

## Conclusion

In this primary obesity prevention intervention, we did not see effects of the intervention in relation to sleep duration, sleep onset latency, parenting stress, or pro-social behavior. Children in the intervention group had more behavioral problems after the intervention, but the difference is not considered clinically meaningful and scores were within normal range both before and after the intervention.

## Supporting information

**S1 Table. Questions selected and modified from the Swedish version of the parenting stress index.**
(PDF)

**S2 Table. Minimal detectable effects for study outcomes.**
(PDF)

**S3 Table. Observations and numbers of missing values for imputed variables.**
(PDF)

**S4 Table. Changes in sleep and stress between baseline and follow-up (1.3 years).**
(PDF)

**S5 Table. Baseline outcomes stratified by completers and non-completers.**
(PDF)

**S1 Appendix. Original study protocol.**
(PDF)

**S2 Appendix. CONSORT checklist.**
(PDF)

## Acknowledgments

The authors would like to thank all the families participating in the Healthy Start intervention. Also, the authors would like to thank the Danish Health Visitor's Child Health Database and

the general practitioners for providing data. Finally, the authors would like to thank all the researchers, project staff, and students that helped design and conduct the Healthy Start intervention study.

## Author Contributions

**Conceptualization:** Nanna Julie Olsen, Berit Lilienthal Heitmann.

**Data curation:** Nanna Julie Olsen, Jeanett Friis Rohde, Maria Stougaard, Mina Nicole Händel, Berit Lilienthal Heitmann.

**Formal analysis:** Sofus Christian Larsen.

**Funding acquisition:** Berit Lilienthal Heitmann.

**Investigation:** Nanna Julie Olsen, Jeanett Friis Rohde, Maria Stougaard, Mina Nicole Händel, Berit Lilienthal Heitmann.

**Methodology:** Nanna Julie Olsen, Sofus Christian Larsen, Jeanett Friis Rohde, Berit Lilienthal Heitmann.

**Project administration:** Nanna Julie Olsen, Berit Lilienthal Heitmann.

**Resources:** Nanna Julie Olsen, Jeanett Friis Rohde, Maria Stougaard, Mina Nicole Händel, Berit Lilienthal Heitmann.

**Software:** Sofus Christian Larsen.

**Supervision:** Berit Lilienthal Heitmann.

**Validation:** Nanna Julie Olsen, Sofus Christian Larsen, Jeanett Friis Rohde, Berit Lilienthal Heitmann.

**Visualization:** Nanna Julie Olsen, Sofus Christian Larsen, Jeanett Friis Rohde, Berit Lilienthal Heitmann.

**Writing – original draft:** Nanna Julie Olsen, Sofus Christian Larsen, Jeanett Friis Rohde, Maria Stougaard, Mina Nicole Händel, Ina Olmer Specht.

**Writing – review & editing:** Nanna Julie Olsen, Sofus Christian Larsen, Jeanett Friis Rohde, Maria Stougaard, Mina Nicole Händel, Ina Olmer Specht, Berit Lilienthal Heitmann.

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
