## [Editor Report · Decision Letter 0]

21 Jun 2021

PONE-D-21-15575

Effects of the Healthy Start randomized intervention on psychological stress and sleep habits among obesity-susceptible healthy weight children and their parents.

PLOS ONE

Dear Dr. Olsen,

Thank you for submitting your manuscript to PLOS ONE. After careful consideration, we feel that it has merit but does not fully meet PLOS ONE’s publication criteria as it currently stands. Therefore, we invite you to submit a revised version of the manuscript that addresses the points raised during the review process.

First let me say that I appreciate that you are working to publish these null findings for this secondary outcome in this trial. I strongly support publishing null findings and PLOS One’s mission makes this a good fit. That said, I believe that the bar is and should be rather high for publishing null results. In particular, the current study is missing any power analyses, readers and reviewers must be convinced that the study had a good chance of finding a clinically meaningful effect if it was there. Therefore, I’m assigning a decision of major revision and asking for edits before I send the paper out for external review. Please note that if you submit a revision, the reviewer’s comments in the next round will determine the next decision. I include further discussion below of some specific comments that I noted in reading the paper.

The data analytic model is not adequately described, while you say you used regression, I don’t know if this was the outcome regressed on treatment status and controlling for baseline, or was it change in the outcome regressed on treatment status? Please document the analysis so anyone could replicate the results. Also, the language around ITT as the analysis using multiple imputation and protocol as the analysis using listwise deletion is somewhat confusing, it might be better to report just one of these (the MI results would be my preference) in the paper and then note the other one.There appears to be substantial drop out which is greater in the treatment condition than the control condition.  I would find this concerning in any intervention, while MI may partially adjust for this it wouldn’t adjust for systematic dropout (some families were unhappy with the treatment, or the children were doing well and so they didn’t want any more contact). This should be explored somewhat more and mentioned as a possible reason for failure to find effects.I’m doing this review off-line and so I don’t have access to the clinical trials registration, but I will read it for the next round and would hope that initial protocols were followed and any deviations explained.Power analyses are required and to the extent that these were originally done for these outcomes, the original power analyses should be presented. If they were not done for these outcomes then new power analyses are needed, paying particular attention to justifying the effect size that is hypothesized for each of the outcomes given other intervention studies, clinically meaningful effects, and noting the reliability of the outcome measures (measures with low reliability would generally have smaller effect sizes). There should be enough detail in the power analyses that I could replicate the results.

We look forward to receiving your revised manuscript.

Kind regards,

Lee Van Horn, PhD

Academic Editor

PLOS ONE

Journal Requirements:

2. Please include captions for your Supporting Information files Protocol and CONSORT checklist at the end of your manuscript, and update any in-text citations to match accordingly. Please see our Supporting Information guidelines for more information: http://journals.plos.org/plosone/s/supporting-information.

3.We note that you have indicated that data from this study are available upon request. PLOS only allows data to be available upon request if there are legal or ethical restrictions on sharing data publicly. For information on unacceptable data access restrictions, please see http://journals.plos.org/plosone/s/data-availability#loc-unacceptable-data-access-restrictions. 
---

## [Author Response · Author response to Decision Letter 0]

27 Aug 2021

Frederiksberg, August 25th, 2021

Manuscript ID PONE-D-21-15575

Dear Editor

Thank you for the comments to our manuscript. The comments are included below and our response to each point is marked in italic. In addition to the comments, we have added some minor typographical and linguistic improvements added to the paper. 

All changes in the revised paper are marked with tracked changes. A clean revision with no tracked changes is also uploaded. Please note that the page- and line numbers below are based on the clean revision. 

We very much appreciate constructive comments and suggestions. 

Editor Comments to Author:

1. The data analytic model is not adequately described, while you say you used regression, I don’t know if this was the outcome regressed on treatment status and controlling for baseline, or was it change in the outcome regressed on treatment status? Please document the analysis so anyone could replicate the results. Also, the language around ITT as the analysis using multiple imputation and protocol as the analysis using listwise deletion is somewhat confusing, it might be better to report just one of these (the MI results would be my preference) in the paper and then note the other one.

The analyses were performed using change in the outcome regressed on treatment status. Based on the comment, the Statistical methods section has been rewritten with the aim of improving the description of the data analytical model. 

Page 11, line 252: “To test the effect of the intervention, linear regression models with treatment status included as the explanatory variable and changes in sleep duration, sleep onset latency, SDQ scores or PSI during follow-up included as the response variables were conducted. All models were adjusted for baseline measure of outcome.”

Table 2, headline: “Changes in sleep and stress between baseline and follow-up (1.3 years).”

Page 11, line 258: “Children who dropped out between baseline and follow-up were included in the analyses according to the intention-to-treat (ITT) principle using multiple imputation. In the multiple imputations, m=10 complete data sets were generated. In each set of data, the missing values were replaced with imputed values, constructed based on predictive distributions for each of the missing values. Each of the completed data sets were analyzed, and the results from the ten analyses were combined to create a single set of estimates that comprised the variability associated with the missing values. The imputations were made using chained equations as implemented in Stata through the commands ice and mim, and based on the variables described above. Imputations were based on allocated group, municipality of birth, baseline BMI z-score, sex, age at baseline, maternal and paternal educational level, perception of the child’s sleep, if the child takes naps during the day, and physical activity compared to children in the same age. When average nighttime sleep duration was used as outcome, imputations were also made on baseline average nighttime sleep duration and SDQ-TD score. When sleep onset latency was used as outcome, imputations were also made on baseline sleep onset latency and SDQ-TD score. When SDQ-TD score was used as outcomes, imputations were also made on baseline SDQ-TD score and average nighttime sleep duration. When SDQ-PSB score was used as outcome, imputations were also made on baseline SDQ-PSB score and average nighttime sleep duration. When PSI score was used as outcome, imputations were also made on baseline PSI score. All statistical tests were two-sided with a significance level at 0.05. Analyses were performed using Stata SE 14 (StataCorp LP, College Station, Texas, USA; www.stata.com).”

Page 12, line 280: “Per protocol analyses, removing data from children who dropped out of the intervention before the 1.3 years follow-up examination, were performed as sensitivity analyses. Per protocol analyses are shown in Supporting S3 table. In addition, possible effect modification by sex was examined in all models.”

Thank you for the suggestion to report just one of the analyses. We have revised the manuscript and now present only the MI results and have moved the protocol analyses to Supporting S3 table. We have rephrased the results sections accordingly and have deleted any text on the protocol analyses (pages 14-16). We have also removed the results from the protocol analyses from Table 2. 

In addition, we have rephrased the abstract and the manuscript text:

Abstract, line 46: “…. and 95% Confidence Intervals (95% CI) were analyzed using linear regression intention-to-treat (n=543 (intervention group n=271, control group n=272)) analyses.”

Abstract, line 51: “Mean change in SDQ-TD was 0.09 points (95% CI -0.57;0.59) in the intervention group, and -0.69 points (95% CI -1.16; -0.23) in the control group (p=0.06).” 

Page 11, line 252: “To test the effect of the intervention, linear regression models with treatment status included as the explanatory variable and changes in sleep duration, sleep onset latency, SDQ scores or PSI during follow-up included as the response variables were conducted. All models were adjusted for baseline measure of outcome.”

Page 12, line 290: “Sample size was n=543”

2. There appears to be substantial drop out which is greater in the treatment condition than the control condition. I would find this concerning in any intervention, while MI may partially adjust for this it wouldn’t adjust for systematic dropout (some families were unhappy with the treatment, or the children were doing well and so they didn’t want any more contact). This should be explored somewhat more and mentioned as a possible reason for failure to find effects.

We have added a table to the manuscript with outcomes stratified by completers and non-completers, showing that non-completers had a statistically significant higher SDQ-TD score than completers, while no other significant differences were observed. 

Page 12, line 283: “Supporting S4 table shows the specific outcomes stratified by completers and non-completers. Non-completers had a statistically significant higher SDQ-TD score than completers, while no other significant differences were observed.”

This has also been addressed in the Discussion section:

Page 17, line 340: “The intervention group had a larger drop-out rate than the control group, which may have introduced a systematic drop-out that is not fully adjusted for in the multiple imputations. Even though we only observed a statistically significant difference between completers and non-completers in SDQ-TD score, this cannot be ruled out as a possible reason for failure to find effects.”

3. I’m doing this review off-line and so I don’t have access to the clinical trials registration, but I will read it for the next round and would hope that initial protocols were followed and any deviations explained.

Page 6, line 139: “The intervention period was shortened from the expected 2 years to 1.3 years, because the planning of practicalities related to the conduction of the study and recruitment of participants took longer than anticipated.”

The original protocol has been uploaded as Supporting S1 Appendix. 

4. Power analyses are required and to the extent that these were originally done for these outcomes, the original power analyses should be presented. If they were not done for these outcomes then new power analyses are needed, paying particular attention to justifying the effect size that is hypothesized for each of the outcomes given other intervention studies, clinically meaningful effects, and noting the reliability of the outcome measures (measures with low reliability would generally have smaller effect sizes). There should be enough detail in the power analyses that I could replicate the results.

Page 10, line 247: “The sample size of the Healthy Start Study was established with the main purpose of having enough statistical power to detect a potential effect of the intervention on the primary outcome (body weight). As the present results represent a secondary study, the sample size was not determined for the purpose of these analyses. However, minimal detectible effects are calculated for all outcomes included in the present study (Supporting S2 table).” 

Page 17, line 352: “Based on the minimal detectable effects from the power calculations for these secondary outcomes, it is considered unlikely that we have missed intervention effects that would be clinically relevant.“

---

## [Decision Letter · Decision Letter 1]

22 Nov 2021

PONE-D-21-15575R1Effects of the Healthy Start randomized intervention on psychological stress and sleep habits among obesity-susceptible healthy weight children and their parents.

PLOS ONE

Dear Dr. Olsen,

Thank you for submitting your manuscript to PLOS ONE. After careful consideration, we feel that it has merit but does not fully meet PLOS ONE’s publication criteria as it currently stands. Therefore, we invite you to submit a revised version of the manuscript that addresses the points raised during the review process.

I now have reviews for your paper from 4 reviewers, 3 of whom had significant comments. Based on the consensus of these reviewers and my own reading of your paper my editorial decision is that a minor revision is still needed before the paper could be published in Plos ONE. Please pay attention to all of the comments, that said, I disagree with reviewer 1 who says that the analyses on detectable differences are not needed since this is a secondary analysis. While there is some controversy over use of post hoc methods for power, the controversy stems from other issues rather than the issue here where you did not find the effect and are making the case for there being no meaningful effect given the study design. I wouldn’t be willing to accept a paper with null findings without a measure of power, without knowing if you could have found a meaningful effect the results are not very useful. A good reference for this is:

Gelman, A., and Carlin, J. B. (2014).  Beyond power calculations:  Assessing Type S (sign) and Type M (magnitude) errors.  Perspectives on Psychological Science 9, 641-651.

Review 3 notes that it might have been odd to expect an effect on sleep given that most of your subjects were in the normal range to begin with. I think a response to this and inclusion of the power analyses might go hand in hand.   

We look forward to receiving your revised manuscript.

Kind regards,

Lee Van Horn, PhD

Academic Editor

PLOS ONE

Journal Requirements:

Additional Editor Comments (if provided):

Reviewers' comments:

Reviewer's Responses to Questions

**Comments to the Author**

1. If the authors have adequately addressed your comments raised in a previous round of review and you feel that this manuscript is now acceptable for publication, you may indicate that here to bypass the “Comments to the Author” section, enter your conflict of interest statement in the “Confidential to Editor” section, and submit your "Accept" recommendation.

Reviewer #1: (No Response)

Reviewer #2: (No Response)

Reviewer #3: (No Response)

Reviewer #4: (No Response)

2. Is the manuscript technically sound, and do the data support the conclusions?

Reviewer #1: Yes

Reviewer #2: Yes

Reviewer #3: Partly

Reviewer #4: Yes

3. Has the statistical analysis been performed appropriately and rigorously? 

Reviewer #1: Yes

Reviewer #2: Yes

Reviewer #3: I Don't Know

Reviewer #4: Yes

4. Have the authors made all data underlying the findings in their manuscript fully available?

Reviewer #1: Yes

Reviewer #2: No

Reviewer #3: Yes

Reviewer #4: Yes

5. Is the manuscript presented in an intelligible fashion and written in standard English?

Reviewer #1: Yes

Reviewer #2: Yes

Reviewer #3: Yes

Reviewer #4: Yes

6. Review Comments to the Author

Reviewer #1: Since this is the secondary analysis, it is not meaningful to calculate the detectable effect size based on the collected sample size. Remove the sentence of Line 250-251 and table S2. You can add the reference to the power for the main study in Line 247-249.

Table 1. Since Wilcoxon was used, better present median with IQR.

Table 2 footnote, add covariates adjusted in the regression model. Did you test normality of the regression, however, nonparametric method was used in Table 1. This needs justification.

Flowchart better use two arms for intervention and control groups.

You can briefly mention the findings from the sensitivity analysis in the Results section.

Reviewer #2: Dear authors and editorial team,

Thank you for the opportunity to review this manuscript. I declare that I do not have any competing interests in completing this role. The aim of this study was to investigate changes to children’s sleep quality and parents’ and children’s perceived stress as secondary outcomes of a family-based obesity prevention randomized controlled trail. The authors report no statistically significant differences at follow-up, but nevertheless, this research adds to our collective understanding of the many factors that shape children’s obesity and chronic disease risk. The manuscript is well-written and the analyses are thorough. I have provided minor suggestions to improve clarity. Thank you for continuing this important work into protecting families’ long-term health and happiness.

Introduction, page 3, lines 62-70: The first sentence provides a good opening, but the remainder of the paragraph seems somewhat abrupt. Your abstract introduction presents the link between children's stress and chronic illness risk upfront, and I would recommend adding a similar sentence or two to this beginning paragraph to underscore the relevance of chronic stress to children's health, then continue with the description of stress assessment strategies. E.g., "... chronic exposure to environmental stressors is common. This has adverse implications for children's health because chronic stress is associated with obesity risk, poor sleep, etc.". Additionally, no capitalization is required after the semicolon in line 64 (“In” to “in”).

Introduction, page 3, line 79: “suggest” should be “suggests”.

Child stress methods, page 8, line 196: I believe you meant “100,000 children” with a comma instead of a period.

Child stress methods, page 9, line 215: capitalize “Swedish”.

Modified intention-to-treat analyses, page 11, lines 268-275: It would be helpful to the reader to contextualize the missing data if the numbers of participants with imputed values were described. This could be added in line where each variable is described, e.g. “Where PSI score was used as an outcome, imputations were also made on baseline PSI score for n = X participants”.

Sensitivity analyses, page 12, line 282: What is meant by “possible effect modification by sex”? Was this tested as moderation with interaction terms, in mediation path analyses, or another method you could describe further?

Discussion, page 19, lines 388-389: I agree that multi-parent responses to the SDQ may have complicated the data. I also wonder if this applies to the PSI results. You mention earlier in the paper that gender of the reporting parent was not assessed, but is it possible to know if the same parent completed both baseline and follow-up PSI surveys? Perceived stress is a highly subjective and gendered concept, and so potential differences between family members or gender differences in stress perception may have also contributed to the observed results. Cooke and colleagues explore this in their paper, albeit for a different family stress scale (DOI: 10.1177/0748175615578756).

Figure 1: Small formatting adjustment needed to read all content in box with “Randomized to allocation group followed in national registers”.

Reviewer #3: Below are my comments:

BACKGROUND/HYPOTHESIS

1. While the references (2&3) supports the level of cortisol reactivity to stress in children predicts internalizing problem in a later age, which illustrates an association between cortisol level and behavioral problem. Nonetheless, behavioral problem in children could be related to a wide range of factors such as parenting skills and child temperament. It does not seem appropriate to assume SDQ is a measurement of child stress. Rather, the authors could make it explicit that this manuscript focuses on the behavioral problems of the participants.

2. As commented by the authors, the baseline sleep duration and SDQ are within normal range, what would be the intended intervention effects that the authors would like to see? Would there be a subgroup of participants who had subthreshold sleep and stress problem that could benefit more from the intervention?

METHOD

1. From the original protocol attached as Appendix 1, the intervention described focused on nutritional counselling and physical activity. Whereas in the introduction, line 97, authors reports that the intervention “focused on improving diet, increasing physical activity, improving sleep duration and quality and reducing psychological stress in the family…”. In that case, the authors would have to elaborate on what intervention was being offered to improve the sleep and psychological stress of the family, so as to allow replication by others.

2. Line 193: for clarity, the authors should explained what consists of the SDQ-TD and SDQ-PSB score.

3. Line 198: for clarity, the authors should state what condition have been validated to use SDQ as a screening tool.

RESULTS

1. Table 1 showed the baseline characteristics of the participants and this should include all participants as illustrate in the flowchart (N=543). In the current version, Table 1 reports only the results of 307 participants, does it mean that data of the remaining 236 participants were already missing at the baseline? In that case, what data were used to compose Suppl Table 4: Baseline outcomes of completers (N=303) and non-completers (N=201)?

DISCUSSION

As the participants in this study has normal sleep and stress parameters to begin with, the intervention from this study is also preventive by nature. The authors may discuss about the trajectory/ emergence on the sleep in child (Williamson A et al J Pediatr 2019), and that this study may not have a long enough follow up to capture the prevention effect. Likewise, the authors may analyze a subgroup of high-risk case with subthreshold sleep problem and see if the intervention effects could be seen.

Reviewer #4: (No Response)

7. PLOS authors have the option to publish the peer review history of their article (what does this mean?). If published, this will include your full peer review and any attached files.

Reviewer #1: No

Reviewer #2: **Yes: **Valerie Hruska

Reviewer #3: No

Reviewer #4: No

---

## [Author Response · Author response to Decision Letter 1]

20 Jan 2022

Frederiksberg, January 20th, 2022

Manuscript ID PONE-D-21-15575R1

Dear Editor

On behalf of all authors, I would like to sincerely thank you and the reviewers for the very thorough and constructive review of our manuscript. 

The comments are included below, and our response to each point is marked in italic. In addition to the comments, we have added some minor typographical and linguistic improvements to the paper. 

All changes in the revised paper are marked with tracked changes. A clean revision with no tracked changes is also uploaded. Please note that the page- and line numbers below are based on the clean revision. 

We very much appreciate the constructive comments and suggestions that we have received. 

Editor comments 

Thank you for submitting your manuscript to PLOS ONE. After careful consideration, we feel that it has merit but does not fully meet PLOS ONE’s publication criteria as it currently stands. Therefore, we invite you to submit a revised version of the manuscript that addresses the points raised during the review process.

I now have reviews for your paper from 4 reviewers, 3 of whom had significant comments. Based on the consensus of these reviewers and my own reading of your paper my editorial decision is that a minor revision is still needed before the paper could be published in Plos ONE. Please pay attention to all of the comments, that said, I disagree with reviewer 1 who says that the analyses on detectable differences are not needed since this is a secondary analysis. While there is some controversy over use of post hoc methods for power, the controversy stems from other issues rather than the issue here where you did not find the effect and are making the case for there being no meaningful effect given the study design. I wouldn’t be willing to accept a paper with null findings without a measure of power, without knowing if you could have found a meaningful effect the results are not very useful. A good reference for this is:

Gelman, A., and Carlin, J. B. (2014). Beyond power calculations: Assessing Type S (sign) and Type M (magnitude) errors. Perspectives on Psychological Science 9, 641-651.

Review 3 notes that it might have been odd to expect an effect on sleep given that most of your subjects were in the normal range to begin with. I think a response to this and inclusion of the power analyses might go hand in hand. 

Thank you very much for the very comprehensive and constructive evaluation of our manuscript. It is greatly appreciated.  

Journal Requirements

We have scrutinized the reference list and have not identified any articles that have been retracted. 

We have made the following changes to the reference list:

Reference 1 has been corrected (publication date updated)

Reference 14 has been corrected (author list updated (American Academy of Sleep Medicine has been spelled out))

Reference 31 in Revision 1 (Knudsen LB, Olsen J. The Danish Medical Birth Registry. Dan Med Bull. 1998;45(3):320-3.) has been replaced with reference 23 in Revision 2 (Bliddal M, Broe A, Pottegard A, Olsen J, Langhoff-Roos J. The Danish Medical Birth Register. Eur J Epidemiol. 2018;33(1):27-36.), as this reference is more new. 

New references have been added according to editor and reviewer comments (reference numbers 2, 3, 27, 34, 36, 47-54)

Additional Editor Comments (if provided):

Review Comments to the Author

Reviewer #1 

Since this is the secondary analysis, it is not meaningful to calculate the detectable effect size based on the collected sample size. Remove the sentence of Line 250-251 and table S2. You can add the reference to the power for the main study in Line 247-249.

We respectfully disagree with this comment, as the results may not be very useful without knowing if meaningful effects could have been detected. 

Line 260-261: “…, as results may not be useful without knowing if meaningful effects could be detected”

Table 1. Since Wilcoxon was used, better present median with IQR.

Thank you for bringing this to our attention. We did not test for differences in baseline characteristics because of the randomized design. The footnote on using Wilcoxon and Chi-squared tests has therefore been deleted.

Table 2 footnote, add covariates adjusted in the regression model. 

Table 2 footnote: “Difference between groups tested using linear regression modeling with information on outcome, group allocation, and baseline measure of outcome.”

Did you test normality of the regression, however, nonparametric method was used in Table 1. This needs justification.

As described above, the footnote on nonparametric tests has been deleted from Table 1. 

Model assumptions were assessed for all models through normal probability plots and residual plots. This has been specified:

Lines 265-266: “Model assumptions (consistency with a normal distribution and variance homogeneity) were assessed for all models through normal probability plots and residual plots.”

Flowchart better use two arms for intervention and control groups.

The flowchart has been updated, using two arms for intervention and control groups

You can briefly mention the findings from the sensitivity analysis in the Results section.

Thank you for this great suggestion. We have added the following sentence in the results section:

Lines 309-310: “Sensitivity analyses showed essentially similar results as the ITT analyses presented below (Supporting S4 table). “

Reviewer #2 

Dear authors and editorial team,

Thank you for the opportunity to review this manuscript. I declare that I do not have any competing interests in completing this role. The aim of this study was to investigate changes to children’s sleep quality and parents’ and children’s perceived stress as secondary outcomes of a family-based obesity prevention randomized controlled trail. The authors report no statistically significant differences at follow-up, but nevertheless, this research adds to our collective understanding of the many factors that shape children’s obesity and chronic disease risk. The manuscript is well-written and the analyses are thorough. I have provided minor suggestions to improve clarity. Thank you for continuing this important work into protecting families’ long-term health and happiness.

Thank you for the kind words on our work. 

Introduction, page 3, lines 62-70: The first sentence provides a good opening, but the remainder of the paragraph seems somewhat abrupt. Your abstract introduction presents the link between children's stress and chronic illness risk upfront, and I would recommend adding a similar sentence or two to this beginning paragraph to underscore the relevance of chronic stress to children's health, then continue with the description of stress assessment strategies. E.g., "... chronic exposure to environmental stressors is common. This has adverse implications for children's health because chronic stress is associated with obesity risk, poor sleep, etc.". 

Thank you for this suggestion. We have added the following sentence to the introduction: 

Lines 60-63: “As chronic stress can initiate inflammatory processes in the body, which can be expressed e.g. in adipose tissue, muscle mass and hormones, stress may have adverse implications for children’s health, and may lead to an increased risk of obesity, metabolic syndrome, and cardiovascular disease”

Additionally, no capitalization is required after the semicolon in line 64 (“In” to “in”).

Corrected

Introduction, page 3, line 79: “suggest” should be “suggests”.

Corrected

Child stress methods, page 8, line 196: I believe you meant “100,000 children” with a comma instead of a period.

Corrected

Child stress methods, page 9, line 215: capitalize “Swedish”.

Corrected

Modified intention-to-treat analyses, page 11, lines 268-275: It would be helpful to the reader to contextualize the missing data if the numbers of participants with imputed values were described. This could be added in line where each variable is described, e.g. “Where PSI score was used as an outcome, imputations were also made on baseline PSI score for n = X participants”.

Thank you for this suggestion. We have added this information to the manuscript and have made a new Supplemental S3 table. 

Lines 289-292: “The number of missing values ranged from 0 for the variables allocated group, municipality of birth, baseline BMI z-score, sex, and age at baseline, to 239 for paternal educational level. Distribution of observed and missing values of each variable included in the imputations are shown in Supporting S3 table.”

Sensitivity analyses, page 12, line 282: What is meant by “possible effect modification by sex”? Was this tested as moderation with interaction terms, in mediation path analyses, or another method you could describe further?

This has now been specified:

Lines 300-302: “In addition, possible effect modification by sex was examined for all outcomes by adding product terms to the models. Subgroup analyses were conducted if statistically significant interactions were observed.”

Lines 310-311: “We found no evidence of effect modification by sex for any of the outcomes (all P> 0.102).”

Discussion, page 19, lines 388-389: I agree that multi-parent responses to the SDQ may have complicated the data. I also wonder if this applies to the PSI results. You mention earlier in the paper that gender of the reporting parent was not assessed, but is it possible to know if the same parent completed both baseline and follow-up PSI surveys? Perceived stress is a highly subjective and gendered concept, and so potential differences between family members or gender differences in stress perception may have also contributed to the observed results. Cooke and colleagues explore this in their paper, albeit for a different family stress scale (DOI: 10.1177/0748175615578756).

Thank you for making this excellent point. We do not have information on whether the PSI surveys were completed by the same parent at baseline and follow-up. We have added this as a limitation in the discussion section:

Lines 415-419: “Likewise, we do not have information on whether the same parent completed the PSI questionnaire at baseline and follow-up. This may have introduced some misclassification, because perceived stress may differ between mothers and fathers, as explored (for a different family stress scale), by Cooke and colleagues. The consequence is an attenuation of the observed differences.” 

Figure 1: Small formatting adjustment needed to read all content in box with “Randomized to allocation group followed in national registers”.

Thank you for bringing this to our attention. Figure 1 has been updated, including the box of the shadow control group. 

Reviewer #3

Below are my comments:

BACKGROUND/HYPOTHESIS

1. While the references (2&3) supports the level of cortisol reactivity to stress in children predicts internalizing problem in a later age, which illustrates an association between cortisol level and behavioral problem. Nonetheless, behavioral problem in children could be related to a wide range of factors such as parenting skills and child temperament. It does not seem appropriate to assume SDQ is a measurement of child stress. Rather, the authors could make it explicit that this manuscript focuses on the behavioral problems of the participants.

Thank you for this constructive comment. We have added the point that behavioral problems may not reflect stress as a limitation in the discussion section:

Lines 419-423: “In our study, child behavioral problems were used as an indicator of child psychological stress. Even though previous studies have found associations between behavioral problems and cortisol, behavioral problems could also be related to factors such as infant temperament and parenting skills. Another limitation to our study is hence that the extent to which SDQ-TD scores reflect stress in the child remains an open question.”

To make it explicit that this manuscript focuses on child behavioral problems, we have made the following adjustments:

Lines 95-96: “The aim of this study was to evaluate the effects of a multifaceted intervention on child behavioral problems, parental psychological stress, as well as on sleep duration and sleep onset latency.”

Line 193: Headline has been changed to “Child behavioral problems”

Line 316: Headline has been changed to “Child behavioral problems”

Lines 194-195: “To indicate child psychological stress, the Danish single-sided version of the Strengths and Difficulties Questionnaire was included in the parental questionnaire.”

2. As commented by the authors, the baseline sleep duration and SDQ are within normal range, what would be the intended intervention effects that the authors would like to see? Would there be a subgroup of participants who had subthreshold sleep and stress problem that could benefit more from the intervention?

Thank you for this excellent question. Considering that the primary aim of the intervention was to prevent overweight development, and that short sleep duration and a high level of stress are considered risk factors for overweight development, the intended intervention effects that we would have liked to see are a longer sleep duration in the intervention group compared to the control group, and a lower SDQ score in the intervention group compared to the control group. It could certainly be hypothesized that there would be a subgroup of participants who had subthreshold sleep and stress problems, where any larger intervention effects may have been diluted. Unfortunately, we do not believe our sample has sufficient statistical power to conduct such analyses. 

METHOD

1. From the original protocol attached as Appendix 1, the intervention described focused on nutritional counselling and physical activity. Whereas in the introduction, line 97, authors reports that the intervention “focused on improving diet, increasing physical activity, improving sleep duration and quality and reducing psychological stress in the family…”. In that case, the authors would have to elaborate on what intervention was being offered to improve the sleep and psychological stress of the family, so as to allow replication by others.

Lines 153-155: “The sleep component of the intervention was focused on sleep hygiene, and recommendations for sleep duration. The stress component of the intervention was focused on spending more time as a family.”

Lines 161-170: “To support the health consultants, and to ensure homogeneity in the delivery of the intervention, detailed descriptions on how to provide the consultations were developed by the health consultants in collaboration with the project management. The content of each session was selected from evidence-based determinants of the lifestyle habits that the Healthy Start intervention study aimed to optimize. For example, a high media consumption has been associated with a shorter sleep duration, and lowering media consumption could therefore increase the sleep duration. The delivery of the sessions was pilot-tested and trained internally by the health consultants and the project management. Detailed descriptions on how each session was provided have been published previously.” 

2. Line 193: for clarity, the authors should explained what consists of the SDQ-TD and SDQ-PSB score.

Lines 198-202: “The scores of the “Emotional symptoms”, “Conduct problems”, “Hyperactivity/inattention”, and “Peer relationship problems” scales are summed to a SDQ-TD score, based on a scoring syntax available from the SDQ-webpage, and used as an exposure variable. The SDQ-TD score ranges from 0-40 points, with higher scores being worse. 

Lines 203-205: “The score of the “Prosocial behavior” scale (SDQ-PSB score) is not incorporated into the SDQ-TD score, as absence of prosocial behaviors differs conceptually from the presence of psychological difficulties”

3. Line 198: for clarity, the authors should state what condition have been validated to use SDQ as a screening tool.

This paragraph has been rewritten:

Lines 206-209: “The SDQ has been completed for nearly 100,000 children and adolescents in both

population studies and clinical samples in the Scandinavian countries. Moreover, in four large-scale Danish cohorts, SDQ has shown to be a useful screening tool for boys and girls across age groups (between 5 and 12 years) and raters (parents and teachers)”

RESULTS

1. Table 1 showed the baseline characteristics of the participants and this should include all participants as illustrate in the flowchart (N=543). In the current version, Table 1 reports only the results of 307 participants, does it mean that data of the remaining 236 participants were already missing at the baseline? In that case, what data were used to compose Suppl Table 4: Baseline outcomes of completers (N=303) and non-completers (N=201)?

Thank you for bringing this to our attention. We have updated Table 1 so it now includes all participants as illustrated in the flowchart (n=543)

DISCUSSION

As the participants in this study has normal sleep and stress parameters to begin with, the intervention from this study is also preventive by nature. The authors may discuss about the trajectory/ emergence on the sleep in child (Williamson A et al J Pediatr 2019), and that this study may not have a long enough follow up to capture the prevention effect. Likewise, the authors may analyze a subgroup of high-risk case with subthreshold sleep problem and see if the intervention effects could be seen.

Thank you for this excellent point. We have added to the discussion section:

Lines 342-346: “Because sleep problems may have different trajectories between infancy and middle childhood, as investigated by Williamson and colleagues, another possible explanation for our results could be that the follow-up period was too short to capture a prevention effect. However, the post hoc power calculations of minimal detectable effects in this study does not suggest that we have missed any clinically relevant intervention effects.”

Reviewer #4

(No Response)

---

## [Editor Report · Decision Letter 2]

14 Feb 2022

Effects of the Healthy Start randomized intervention on psychological stress and sleep habits among obesity-susceptible healthy weight children and their parents.

PONE-D-21-15575R2

Dear Dr. Olsen,

Thank you for your work in addressing the concerns raised by the reviewers and myself. After carefully reading the manuscript I'm pleased to inform you that your manuscript has been judged scientifically suitable for publication and will be formally accepted for publication once it meets all outstanding technical requirements.

Kind regards,

Lee Van Horn, PhD

Academic Editor

PLOS ONE
---

## [Editor Report · Acceptance letter]

28 Feb 2022

PONE-D-21-15575R2 

Effects of the Healthy Start randomized intervention on psychological stress and sleep habits among obesity-susceptible healthy weight children and their parents. 

Dear Dr. Olsen:

I'm pleased to inform you that your manuscript has been deemed suitable for publication in PLOS ONE. Congratulations! Your manuscript is now with our production department. 

Kind regards, 

on behalf of

Dr. Lee Van Horn 

Academic Editor

PLOS ONE